Complete chloroplast genome studies of different apple varieties indicated the origin of modern cultivated apples from Malus sieversii and Malus sylvestris

Li Xueli
Ding Zhijie
Miao Haoyu
Bao Jinbo
Tian Xinmin tianxm06@lzu.edu.cn
Xinjiang Key Laboratory of Biological Resources and Genetic Engineering, College of Life science and Technology, Xinjiang University , Xinjiang, Urumqi , China
Uversky Vladimir
Electronic publication date: 2022 Mar 18
Publication date: 2022
Volume: 10
Electronic Location ID: e13107
Received 2021 Oct 21; Accepted 2022 Feb 22
Copyright: © 2022 Li et al.
Copyright year: 2022
Copyright holder: Li et al.
License: This is an open access article distributed under the terms of the Creative Commons Attribution License, which permits unrestricted use, distribution, reproduction and adaptation in any medium and for any purpose provided that it is properly attributed. For attribution, the original author(s), title, publication source (PeerJ) and either DOI or URL of the article must be cited.
License URL: https://creativecommons.org/licenses/by/4.0/

Keywords: Apple, Chloroplast genome, Structure variation, Phylogenetic analysis

Funding: Open project of Xinjiang Key Laboratory of Biological Resources and Genetic Engineering 2020D04033 This study was supported by the Open project of Xinjiang Key Laboratory of Biological Resources and Genetic Engineering (grant 2020D04033 to Xinmin Tian). The funders had no role in study design, data collection and analysis, decision to publish, or preparation of the manuscript.

==============================
Background

Apple is one of the most important temperate deciduous fruit trees worldwide, with a wide range of cultivation. In this study, we assessed the variations and phylogenetic relationships between the complete chloroplast genomes of wild and cultivated apples (Malus spp.).

Method

We obtained the complete chloroplast genomes of 24 apple varieties using next-generation sequencing technology and compared them with genomes of (downloaded from NCBI) the wild species.

Result

The chloroplast genome of Malus is highly conserved, with a genome length of 160,067–160,290 bp, and all have a double-stranded circular tetrad structure. The gene content and sequences of genomes of wild species and cultivated apple were almost the same, but several mutation hotspot regions (psbI-atpA, psbM-psbD, and ndhC-atpE) were detected in these genomes. These regions can provide valuable information for solving specific molecular markers in taxonomic research. Phylogenetic analysis revealed that Malus formed a new clade and four cultivated varieties clustered into a branch with M. sylvestris and M. sieversii, which indicated that M. sylvestris and M. sieversii were the ancestor species of the cultivated apple.

Introduction

Malus is one of the most important economic fruit crops in temperate regions. It is composed of approximately 30–35 deciduous trees or shrubs of Rosaceae (Giomaro et al., 2014). Over thousands of years of evolution, thousands of excellent apple varieties have been produced (Morgan et al., 2002), such as Red Delicious, Golden Delicious, Ralls, and Red Fuji. They are becoming increasingly popular worldwide because of their good taste, nutrient-rich value, storability, and convenience. In terms of apple production, China is currently the largest apple producer in the world, with apple planting area and output accounting for more than 50% of the world (Li et al., 2021). In addition to their economic value, apples play an important role in preventing diseases, such as Parkinson’s disease and all kinds of cancer, as well as reducing the risk of diabetes and lowering cholesterol (Rupasinghe, Thilakarathna & Nair, 2013). Therefore, the improvement of apple varieties and cultivation of new varieties is particularly important. The essential questions of the origin and evolution of cultivated species ultimately arise from the identification of wild ancestral populations. Studying the relationship between cultivated apples and their possible predecessors is important for studying the origin of cultivated apples. Additionally, the time, place, and mode of origin of cultivated apples are also the core of the study on the origin of cultivated apples.

To better understand the origin of cultivated apples and their relationship with their major wild ancestors, researchers have conducted numerous studies on the origin of cultivated apples and the phylogenetic relationships between cultivated apples and different wild apples. In 1926, Nikolai Vavilov suggested that the wild species and their related species in Central Asia were the ancestors of the modern cultivated apple, and the whole process of apple cultivation and domestication can be traced back to the Almaty region of Kazakhstan (Vavilov, 1926). However, the only wild species in Central Asia is Malus sieversii Roem. Therefore, the wild species referred to by predecessors (Harris, Robinson & Juniper, 2002) should be M. sieversii, which still exists in Central Asia today. Later, Li (1989) found that M. sieversii is the ancestor of cultivated apples through the investigation of wild apples in Central Asia and believed that Ili in Xinjiang is the origin centre of M. sieversii and the source of diversified cultivated apples, whereas Velasco et al. (2010) and Cornille et al. (2012) contended that the primary and secondary contributors of cultivated varieties are M. sieversii and M. sylvestris, respectively. Sun et al. (2020) proved that the two wild ancestors made great genetic contributions to cultivated apples by analysing the genome and the re-sequencing data of cultivated apples and wild apples (M. sieversii and M. sylvestris). It also showed that M. sylvestris and M. sieversii were the common ancestors of cultivated apples. In summary, researchers have conducted extensive research on the origin of cultivated apples and the phylogenetic relationship between cultivated apples and different wild apples. However, in previous studies on the evolution of apple populations, cultivated apples were usually treated as a group while focusing on the genetic relationship between cultivated apples and different wild apples. Almost no one has studied the genetic differences between different cultivated apples, and there are few studies on the population history of apples. With the development of sequencing technologies, biological science research has entered the era of big data. Genome-level traceability and homology analysis can address a number of important scientific questions, such as crop origin and domestication. At the same time, we can also detect the changes that have occurred in the genome structure and sequence of crops during domestication. However, most of the research on genetic breeding and improvement of crops is focused on the nuclear genome and rarely on the organellar genome.

The nucleus, chloroplast and mitochondria are the three main organelles in the cell that contain genomes. They play an important role in plant activities (Yin et al., 2018). Since Ohyama et al. (1986) first obtained the chloroplast genome of Nicotiana tabacum in 1986, this is the first study to observe and study the structure and characteristics of the chloroplast genome at a more micro level, which is of great significance to the in-depth study of the chloroplast genome. A typical chloroplast genome structure consists of four stable parts: two reverse repeat regions, which are separated by a large single-copy (LSC) region, and a small single-copy (SSC) region (Raubeson & Jansen, 2005). Angiosperm chloroplast genomes generally contain 110–130 genes, most of which are protein-coding genes (involving photosynthetic reaction and gene expression), and the rest encode tRNA and rRNA (Green, 2011). Compared with the other two organellar genomes, chloroplast DNA has several advantages, such as maternal inheritance, multiple copies and a simple structure, and chloroplast genome is highly conserved in both gene content and order. In addition, complete chloroplast genome analysis can provide more genetic information than gene fragments. Therefore, chloroplast genome sequencing has a significant contribution to the species identification and phylogeny of angiosperms (Xi et al., 2012; Xie et al., 2019). For example, Nikiforova et al. (2013) showed the genetic relationship between wild apple (M. sieversii and M. sylvestris) and cultivated apple through the phylogenetic analysis of 47 chloroplast genomes and clarified the contribution of wild species to the maternally inherited genome of domesticated species. Shen et al. (2017) revealed a sister relationship between different Asteraceae species through a comparative analysis of chloroplast genome sequences of five Asteraceae species.

Crop domestication is a process of artificial selection that promotes interdependence between humans and crop plants (Duan et al., 2017). The domestication of apples as cultivars has occurred over thousands of years. Studies have shown that crops accumulate some variation in their cellular DNA during domestication and cultivation (Schmid-Siegert et al., 2017). Chloroplasts, the organelles of cellular photosynthesis, also have their own circular DNA, which plays a pivotal role in the life activities of plants. Therefore, starting from the chloroplast genome, it was further demonstrated that M. sieversii and M. sylvestris are the ancestral species of the four cultivated apples. The following scientific problems are yet to be completely solved: Did the apples also accumulate some variation in their organellar DNA during domestication and cultivation mainly through clonal reproduction (grafting)? What is the mutation rate? What structural variants occur in the genome? What structural variations have taken place in the genome? In this study, 24 genomic sequences of four different Malus germplasms were sequenced. DNA variation was detected to reveal the differences in chloroplast genomes between the two progenitor species and cultivated apples of different varieties. Also, a phylogenetic tree was reconstructed using the complete chloroplast genome to show the phylogenetic relationships among modern cultivated apples, M. sieversii and M. sylvestris. This study provides a strong basis for apple evolution and domestication and an important reference for variety breeding and parental selection in the development of the apple industry.

Materials and Methods

Sampling and DNA extraction

In this study, 24 samples of four representative Malus varieties (Red Delicious, Golden Delicious, Red Fuji, and Ralls) were collected from different provinces in China. All samples were immediately frozen with silica gel and stored at −20 °C (Yang et al., 2020). Red Fuji varieties were selected from three provenances with three individuals in each region; for other cultivars, three different provenances were selected for each cultivar (Table 1). We extracted DNA from 24 samples using the new Tiangen plant genomic DNA extraction kit (Tiangen, Urumqi, Xinjiang, China).

Table 1 Apple varieties and quantity in different places.

Varieties	Source	Longitude
(E)	Latitude
(N)	Number	
Red Delicious	Wunan Town, Wuwei City, China	102.73°	37.82°	2	
Red Delicious	Xingcheng, Liaoning, China	120.72°	40.61°	3	
Red Delicious	Zhengzhou Fruit Tree Institute, China	113.64°	34.75°	1	
Golden Delicious	Wunan Town, Wuwei City, China	102.73°	37.82°	1	
Golden Delicious	Xingcheng, Liaoning, China	120.72°	40.61°	3	
Golden Delicious	Zhengzhou Fruit Tree Institute, China	113.64°	34.75°	1	
Ralls	Xingcheng, Liaoning, China	120.72°	40.61°	3	
Ralls	Zhengzhou Fruit Tree Institute, China	113.64°	34.75°	1	
Red Fuji	Wunan Town, Wuwei City, China	102.73°	37.82°	3	
Red Fuji	Zhengzhou Fruit Tree Institute, China	105.72°	34.58°	3	
Red Fuji	Xingcheng, Liaoning, China	120.72°	40.61	3	

Chloroplast genome sequencing, assembly and annotation

The complete chloroplast genome of 24 cultivated apple accessions were sequenced using the next-generation sequencing. The tested DNA samples were randomly interrupted by Covaris ultrasonic crusher, and the whole library preparation was completed via end repair, addition of polyA-tail, addition of sequencing adapters, purification, PCR amplification. The constructed libraries were sequenced using the Illumina MiSeq 2,000 platform. After they passed the library inspection, the different libraries were pooled according to the effective concentration and target downstream data volume and then sequenced using Illumina HiSeq/MiSeq. The raw image data files obtained from high-throughput sequencing were converted into raw sequences (sequenced reads) by CASAVA base identification analysis, which we called raw data or raw reads, and the results were stored in FASTQ file format, which contains sequence information (reads) and their corresponding sequencing quality information. For each sample, approximately 50.0 GB of raw data were generated, which were assembled using the programme NOVOPlasty (Dierckxsens, Patrick & Guillaume, 2017) with M. sieversii (MH890570.1) as the reference. The focus of the assembly is on the NOVOPlasty configuration file, also known as config.txt; the settings included config.txt parameter values, K-mer = 39, The reference sequence was uploaded; forward reads, and reverse reads uploaded that need to be assembled. The best quality of the assembly appeared as a circular assembly file; however, in general, there are two options, both of which are correct owing to the indeterminate gene order of the two reverse repeat regions of the chloroplast. When one wants to determine which one is correct, sequence comparisons or gene annotations must be performed. All annotated genes were manually curated using Sequin. 16.0 (Wyman, Jansen & Boore, 2004), and start/stop codons and intron/exon boundaries were adjusted using Geneious v4.8.5. Finally, we used the online OGDRAW software to generate a circular genome map (Lohse, Drechsel & Bock, 2007).

Characterization of chloroplast genomes in Malus

The total length of the genome and the length of each region, including LSC, SSC, and inverted repeats (IRs; a pair of reverse complementary repeat regions), gene composition (protein-coding genes, transcriptional RNA genes, ribosomal RNA genes, introns, and exons), base composition, and GC (AT) content were analysed using the Geneious v4.8.5 software.

Analysis of the LSC, SSC, and IR border regions

The IRscope (http://irscope.shinyapps.io/irapp/) online analysis software was used to upload the manually annotated Genbank file and generate a boundary map (Ali, Jaakko & Peter, 2018). Finally, the length differences and related genes of the four regions in different apple varieties were analysed.

Repeat sequence and simple sequence repeats (SSRs)

Four repeats of the chloroplast genome, which included repeats, forward repeats, complement repeats, and palindromic repeats, were identified using the REPuter (https://bibiserv.cebitec.unibielefeld.de/reputer) online software (Kurtz et al., 2001). The setting parameters were as follows: minimum sequence length of 30 bp, edit distances of three bp, minimum alignment score of 50, and sequence identity > 90%. Finally, repetitive sequences of the chloroplast genome were obtained.

MISA (https://webblast.ipk-gatersleben.de/misa/index.php) software was used to detect SSR loci in the genome sequences of wild and cultivated apples (Biswas et al., 2011). The repeat units are mono-, di-, tri-, tetra-, penta-, and hexa-nucleotides. Default: definition (unit_size, min_repeats): 1-10 2-6 3-5 4-5 5-5 6-5; interruptions (max_difference_for_2_SSRs): 100.

Codon usage analysis

All protein-coding genes were used for determining the codon usage. We further analyzed the codon usage frequency and relative synonymous codon usage (RSCU) based on sequences of 83 protein-coding genes in the Malus chloroplast genomes. Avoiding the influence of the amino acid composition, we examined the RSCU using MEGA v7.0, and the results are presented as charts (Sharp & Li, 1987).

Variation analysis

Although chloroplast genomes are highly conserved with regard to composition and sequence, there are some internal mutations and structural variations among the chloroplast genomes of different varieties. In this study, the online software mVISTA (http://genome.lbl.gov/vista/index.shtml) was used for sequence alignment and variation analysis of the apple chloroplast genome (Frazer et al., 2004). The degree of variation in the complete sequence can be evaluated by comparing the similarities between the coding region, non-coding region, introns, and exons of the input sequence with the other sequences. To identify SNPs or short insertion/deletions (InDels), we used the Geneious. v4.8.5 software.

Phylogenetic analysis

To study the phylogenetic position of the chloroplast genomes of cultivated apples and other genera of the Rosaceae family, phylogenetic analysis was performed based on the chloroplast genome sequences for four cultivated apples and 16 Rosaceae species obtained from NCBI (National Center for Biotechnology Information (nih.gov))—M. hupehensis (MK020147.1), M. prunifolia (KU851961.1), M. baccata (KX499859.1), M. micromalus (MF062434.1), M. sylvestris (MK434924.1), M. sieversii (MK434920.1), M. yunnanensis (NC_039624.1), M. florentina (NC_035625.1), M. trilobata (NC_035671.1), Pyrus hopeiensi (MF521826.1), Pyrus ussuriensis (MF521826.1), Pyrus pyrifolia (NC_015996.1), Crataegus kansuensis (MF784433.1), Prunus pedunculata (MG869261.1), Prunus persica (HQ336405.1), and Prunus yedoensis (KU985054.1). We aligned the 20 corresponding sequences using MAFFT 7.017 and adjusted them manually (Katoh et al., 2005). The complete chloroplast genome sequences were used to construct the phylogenetic topology using Bayesian inference (BI) (Huelsenbeck, 2012). Based on the results of the phylogenetic analysis, the phylogenetic relationships between different apple germplasms were further clarified.

Results

Features of the chloroplast genome

The chloroplast genome structure and length of four apple cultivars and two wild ancestors were compared, and the results are summarised in Table 2. The complete chloroplast genome maps of M. domestica and wild apples as shown in Fig. 1. Subtle differences were observed among the chloroplast genome sequences. The results showed that the length of the chloroplast genome of four apple cultivars and two wild ancestors was approximately 160 kb. Among them, the genomes of Red Delicious and M. sieversii were the longest, with their lengths ranging from 160,225–160,290 bp, while those of others were shorter, with lengths ranging 160,067–160,069 bp. The chloroplast genome structures consisted of four stable parts: two IR regions (26,323–26,337 bp), which were separated by an LSC region (88,240–88,440 bp) and an SSC region (19,174–19,340 bp).

Figure 1 Chloroplast genome map of Malus.

Genes lying outside the circle are transcribed in the counter clockwise direction, while those inside are transcribed in the clockwise direction. The colored bars indicate difffferent functional groups. The darker gray area in the inner circle denotes GC content while the lighter gray corresponds to the AT content of the genome. LSC, large single copy; SSC, Small single copy; IR, inverted repeat.

Table 2 Summary of major characteristics of plastomes in Malus.

Varieties	Genome
size (bp)	LSC length (bp)	SSC length (bp)	IR length (bp)	GC (%)	Number of genes	
Total	CDS	tRNAs	rRNAs	infA	ycf1	
Malus sieversii	160,225	88,336	19,179	26,355	36.5	128	83	35	8	1	1	
Malus sylvestris	160,068	88,184	19,180	26,352	36.6	128	83	35	8	1	1	
WW-Red Delicious	160,288	88,440	19,174	26,337	36.5	130	84	37	8	0	1	
160,290	88,440	19,176	26,337	36.5	130	84	37	8	0	1	
LX-Red Delicious	160,288	88,440	19,176	26,337	36.5	130	84	37	8	0	1	
160,290	88,440	19,176	26,337	36.5	130	84	37	8	0	1	
160,290	88,440	19,176	26,337	36.5	130	84	37	8	0	1	
ZZ-Red Delicious	160,290	88,440	19,176	26,337	36.5	130	84	37	8	0	1	
WW-Golden Delicious	160,290	88,440	19,176	26,337	36.5	130	84	37	8	0	1	
LX-Golden Delicious	160,068	88,241	19,181	26,323	36.6	130	84	37	8	0	1	
160,069	88,242	19,181	26,323	36.6	130	84	37	8	0	1	
160,069	88,242	19,181	26,323	36.6	130	84	37	8	0	1	
ZZ-Golden Delicious	160,067	88,240	19,181	26,323	36.6	130	84	37	8	0	1	
LX-Ralls	160,228	88,242	19,340	26,323	36.6	130	84	37	8	0	1	
160,068	88,241	19,181	26,323	36.6	130	84	37	8	0	1	
160,068	88,241	19,181	26,323	36.6	130	84	37	8	0	1	
ZZ-Ralls	160,155	88,242	19,259	26,327	36.6	130	84	37	8	0	1	
WW-Red Fuji	160,068	88,241	19,181	26,323	36.6	130	84	37	8	0	1	
160,069	88,242	19,181	26,323	36.6	130	84	37	8	0	1	
160,069	88,242	19,181	26,323	36.6	130	84	37	8	0	1	
LX-Red Fuji	160,069	88,242	19,181	26,323	36.6	130	84	37	8	0	1	
160,069	88,242	19,181	26,323	36.6	130	84	37	8	0	1	
160,068	88,241	19,181	26,323	36.6	130	84	37	8	0	1	
ZZ-Red Fuji	160,069	88,242	19,181	26,323	36.6	130	84	37	8	0	1	
160,069	88,242	19,181	26,323	36.6	130	84	37	8	0	1	
160,069	88,242	19,181	26,323	36.6	130	84	37	8	0	1	
Note:

WW, Wunan Town, Wuwei City; LX, Xingcheng, Liaoning; ZZ, Zhengzhou Fruit Tree Institute; LSC, large single-copy region; SSC, small single-copy; IR, inverted repeats region; CDS, coding sequence genes.

M. sieversii and M. sylvestris contained 128 genes, including 83 coding sequence (CDS), 35 tRNA, 8 rRNA, and infA and ycf1 genes. Among the 130 genes annotated in cultivated apple varieties, there were 84 CDS, 8 rRNA, 37 tRNA, and ycf1 genes. The LSC region of M. sieversii and M. sylvestris contained 83 genes, accounting for 64.8% of the chloroplast genome, including 21 tRNA genes, 61 protein-coding genes, and infA gene, whereas the SSC region contained 11 CDS genes, accounting for 8.6% of the chloroplast genome. Eight rRNA genes, 12 CDS genes, and 14 tRNA genes were located in the IR region, accounting for 26.6% of the genome. The LSC region of 24 cultivated apples contained 61 CDS genes and 22 tRNA genes, while 11 CDS genes and 1 tRNA gene were located in the SSC region. All eight rRNA genes, 13 CDS genes, and 13 tRNA genes were located in the IR region. The CDS genes in the LSC, SSC, and IR regions accounted for 64.1%, 9.9%, and 26.0% of the total chloroplast genome, respectively. The GC content of the complete chloroplast genome of Malus ranged from 36.5% to 36.6%, with an average content of 36.56%. Specifically, the GC content in the IR region (42.7%) was higher than that in the LSC region (34.1–34.2%) and SSC region (30.3–30.4%) (Table S1).

All genes were divided into three categories according to their functions: photosynthesis-related genes, transcription and translation-related genes, and other genes (Table 3). Among the protein-coding genes of cultivated apples and wild ancestors, 20 genes contained one intron each (trnI-GAU, rps12, ndhB, trnK-UUU, rpL2, rps16, trnG-UCC, atpF, rpoC1, trnL-UAA, trnV-UAC, petB, petD, rpl16, trnI-GAU, trnA-UGC, ndhA, rps12, ndhB, rpL12, and trnA-UGC), and two (clpP and ycf3) had two introns each. The gene with the largest intron (2,497 bp) was trnK-UUU, and the matK gene was included in this gene.

Table 3 A list of genes found in the chloroplast genomes of Malus species.

Function of genes	Group of genes	Gene names	
Photosynthesis	Rubisco	rbcL	
Photosystem I	psaA, psaB, psaC, psaL, psaJ	
Photosystem II	psbA, psbK, psbL, psbD, psbC, psbZ, psbJ, psbF, psbE, psbT, psbN, psbH, psbM	
ATP synthase	atpA, atpFa, atpL, atpE, atpB	
Cytochrome b6/f complex	petN, petA, petL, petG, petDa, petBa	
NADH dehydrogenase	ndhJ, ndhK, ndhC, ndhAa, ndhG, ndhE, ndhD, ndhBa(2), ndhF, ndhH	
Transcription and
translation	RNA polymerase	rpoC1a, rpoC2, rpoA, rpoB	
Ribosomal proteins	rps16a, rps2, rps14, rps4, rps12a, rps11, rps8, rps7(2), rps15, rpL33, rpL20, rpL36, rpL14, rpL16, rpL22, rpL2a(2), rpL23(2)	
Ribosomal RNA	rrn16(2), rrn4.5(2), rrn5(2), rrn23(2)	
transfer RNA	trnQ-UUG, trnS-GCU, trnG-UCCa, trnR-UCU, trnC-GCA, trnD-GUC, trnH-GUG, trnK-UUUa, trnE-UUC, trnT-GGU, trnS-UGA, trnfM-CAU, trnI-GAUa(2), trnA-UGCa(2), trnR-ACG, trnN-GUU, trnL-UAG, trnR-ACG, trnL-CAA, trnL-CAU, trnS-GGA, trnT-UGC, trnL-UAAa, trnV-UACa, trnM-CAU, trnW-CCA, trnP-UGG, trnL-CAU, trnV-GAC	
Other genes	RNA processing	matK	
carbon metabolism	cemA	
fatty acid synthesis	accD	
cytochrome synthesiz gene	ccsA	
	ATP-dependent protease subunit gene	clpPb	
Unknown	conserved reading frames	ycf1, ycf2(2), ycf3b, ycf4	
Notes:

a contains one intron.

b contains two introns, (2) shows genes duplicated in the IR regions.

IR expansion and contraction

Different species have different gene sequences in the four junction regions, and the contraction or expansion of the IR region usually leads to length changes in the chloroplast genome. To further analyse the boundary differences between the two wild species and cultivated apple varieties, four apple samples from different varieties were selected. It can be seen from the results that structural variation was still found in the IR-SSC and IR-LSC boundary in Malus. The IR regions of M. sieversii, M. sylvestris, Red Delicious, Golden Delicious, Red Fuji, and Ralls are shown in Fig. 2. The rps19 gene had an extension of 85 bp in the IRa region and 114 bp in M. sieversii and M. sylvestris. Because the ycfl gene was located at the boundary between SSC and IRb, a 1,074 bp long pseudogene fragment was generated in the IRa region. The ndhF gene crossed the IRa/SSC boundary and was located in the IRa region with a length spanning 12 bp. In addition, the trnH-GUG genes in the four cultivated apple varieties and two wild species were all located in the LSC region, and its distance from the IRb/LSC boundary was between 32 and 82 bp. In general, the genus Malus showed similar characteristics in the IR/SC boundary region.

Figure 2 Comparisons of LSC, SSC and IR region borders among for Malus chloroplast genomes.

Genes are denoted by colored boxes. The number above the gene features shows the distance between the end of the gene and the borders sites. The slashes indicate the location of the distance.

Repeat sequence and SSR analyses

In this study, Malus-related sequences (n = 24) were narrowed to 11 representative sequences and two progenitor sequences for comparative analysis. In these chloroplast genomes, the number of the four repeat types was similar, and their overall distribution in the chloroplast genome was highly conserved. A total of 633 repeats were detected in the 11 chloroplast genomes, including 337, 220, 60, and 16 forward repeats, palindromic repeats, reverse repeats, and complementary repeats, respectively (Fig. 3A), and four repeats were observed in 52.8%, 35.3%, 9.9%, and 2%, respectively (Fig. 3B). The length of the repeat sequence was 30–65 bp and mainly 30–40 bp among Malus (Fig. 3C). Most of these repeats were located in the intergenic spacer and gene region, accounting for 66% and 27.1%, respectively, while only 6.9% were located in the intron region (Fig. 3D).

Figure 3 Investigation of repeated sequences in Malus chloroplast genomes.

(A) The number of different repeat types: P, palindrome repeat; F, forward repeat; R, reverse repeat; C, complementary repeat. (B) Proportion of different repeat types. (C) Number of the repeats by length. (D) Proportion of repeats in Intron, CDS and Spacer regions. WW, Wunan Town, Wuwei City; LX, Xingcheng, Liaoning; ZZ, Zhengzhou Fruit Tree Institute; RD, Red Delicious; GD, Golden Delicious; si, M. sieversii; syl, M. sylvestris.

The types, distribution and number of SSRs in the chloroplast genome of Malus were counted (Table S2). A total of 2,279 SSRs of single nucleotides, dinucleotides, and complex polynucleotides were detected in the chloroplast genome of Malus. Two wild and 11 cultivated apple sequences were selected. The number of mononucleotide, dinucleotide, and complex nucleotide repeats ranged 45–60, 2–3, and 8–9, respectively, in four cultivated apples and two progenitor speices (Fig. 4A). The most common mononucleotide repeats were A/T repeats. SSRs were broadly similar across genomes, compared with SSC and IR regions, most of which were located in the LSC region, and most SSRs were located in the intergenic spacer region (Fig. 4B, Table S3).

Figure 4 Distribution and numbers of small sequence repeats (SSRs) in the chloroplast genomes of Malus.

(A) Total number of repeat; (B) number of repeats in LSC, SSC and IR. (WW, Wunan Town, Wuwei City; LX, Xingcheng, Liaoning; ZZ, Zhengzhou Fruit Tree Institute; RD, Red Delicious; GD, Golden Delicious).

Codon usage

A total of 61 codons encoding 20 amino acids were detected (Fig. 5). The minimum value of Relative synonymous codon usage (RSCU) was 0.47 and the maximum value was 1.85. AGA had the highest RSCU value, ranging from 1.85 to 1.97. Methionine and tryptophan were the least common amino acids, whereas leucine, arginine, and serine were the most common amino acids. The RSCU values of 33 and 34 codons exceeded one for M. sieversii and M. sylvestris, respectively; 31 codons of Red Delicious exhibited greater preference (RSCU > 1), and 32 codons of Golden Delicious, Red Fuji, and Ralls showed greater preference (RSCU < 1). Half of the frequently used codons had RSCU value > 1, and all codons ended in A/U, which was consistent with the rich A/T characteristics of the angiosperm chloroplast genome (Table S4).

Figure 5 Codon content for the 20 amino acids and stop codons in 84 protein-coding genes in the four Malus varieties chloroplast genomes.

RSCU, relative synonymous codon usage; Phe, phenylalanine; Leu, leucine; Ile, isoleucine; Met, methionine; Val, valine; Ser: serine; Pro, proline; Thr, threonine; Ala, alanine; Tyr, tyrosine; His, histidine; Gln, glutamine; Asn, asparagine; Lys, lysine; Asp, aspartic acid; Glu, glutamic; Cys, cysteine; Trp, tryptophan; Arg, arginine; Gly, glycine.

Comparative chloroplast genomic analysis

Considering the chloroplast genome sequence of M. sieversii as reference, mVISTA was used to compare the differences in the chloroplast genome sequences of 13 Malus materials. The results are shown in Fig. 6. The whole sequences of 13 chloroplast genomes were highly similar. The sequence variation was mainly concentrated in the non-coding region, and the non-coding region was less conserved than the coding region. In general, the intergenic region was a highly divergent region among these chloroplast genomes, such as psbI-atpA, psbM-psbD, and ndhC-atpE.

Figure 6 Visualization of alignment of Malus chloroplast genome sequences.

The x-axis represents the coordinate in the chloroplast genome. Annotated genes are displayed along the top. The sequences similarity of the aligned regions is shown as horizontal bars indicating the average percent identity between 50% and 100%.

In the four-part structure, 135 SNP sites and 107 indel sites were identified (Table S5, S6). Among all SNP sites, the number of highly variable regions located in the IR region was the lowest, and among 26 InDels, large InDels (>50 bp) were found in the trnT-trnL, psaI-ycf4, rpoB-trnC-GCA, trnR-UCU-atpA, psbA-trnK intergenic spacer and trnG-GCC, trnV-UAC, clpP, and trnV-UAC introns.

Phylogenetic analyses

Phylogenetic analyses revealed strong bootstrap support for each node. All Malus species formed a branch with strong guidance support that the four cultivated apples had a close phylogenetic relationship with M. sieversii and M. sylvestris, in which the species M. sieversii and Red Delicious formed a cluster, while the Golden Delicious, Red Fuji, Ralls and M. sylvestris clustered into another branch (Fig. 7). Therefore, M. sieversii was not the unique parent source of the modern cultivated apple; however, it was a major progenitor.

Figure 7 Optimal phylogenetic tree resulting from analyses of 19 complete chloroplast genomes of Malus and 1 outgroups using Bayesian inference (BI). Support values are Bayesian posterior probability.

Discussion

In previous studies on Rosaceae fruit trees, chloroplast gene numbers ranged from 110 to 130 (Daniell et al., 2016). In this study, we obtained the chloroplast genome sequences of M. sieversii and M. sylvestris from the NCBI database and assembled complete chloroplast genomes of four varieties of Malus species. Sequence analysis revealed that the two progenitors had 128 genes and four M. domestica contained 130 genes. Similar to the other higher plants, the chloroplast genome had a highly conserved structure and size; it showed a typical circular DNA structure and usual length of the chloroplast genome sequence, ranging from 160,067 to 160,290 bp in Malus.

In the chloroplast genome, the IR region is the most conserved region among all the regions. The change in chloroplast genome size is related to the expansion and contraction of IRs, and its evolution is very different (Shen et al., 2017; Ravi et al., 2008). To a certain degree, the variation and evolution of the chloroplast genome is related to its contraction and expansion (Yang et al., 2010). From the perspective of terrestrial plant evolution, the IR region tends to expand; there are subtle differences in gene contraction and expansion among species, and the difference between the maximum and minimum length of the IR region in Malus is 32 bp (Fig. 2). The IR expansion/contraction of the genus showed no remarkable phylogenetic significance.

Codon usage bias is believed to occur due to the differential abundance of tRNAs corresponding to different codons in cells. This phenomenon is of great significance because it is related to the translation of DNA and the synthesis of biologically functional proteins (Akashi, 1997; Bulmer, 1991; Zhang et al., 2012; Wong et al., 2002; Ermolaeva, 2001). The codon usage analysis of chloroplast genomes of two progenitors and four apple varieties revealed that among the encoded amino acids, leucine and tryptophan were the most and least abundant amino acids, respectively, implying that the usage bias of codons is uneven. The chloroplast genome of Malus prefers to use codons with A or T at the third position. This phenomenon is also found in the chloroplast genomes of other angiosperms, such as upland cotton, wheat, and gramineous plants (Gao et al., 2017; Clegg et al., 1994; Mader et al., 2018; Meng et al., 2018). This also proves that the AT content in the chloroplast genome of Malus is slightly higher than the GC content.

SSRs are valuable molecular markers with a high degree of variation within species; they are widely used in plant population genetics, polymorphism investigations, and evolutionary research (Powell et al., 1995). Most of the SSRs identified in the chloroplast genomes of four apple varieties and two ancestral species were found in non-coding regions (Table S3), which was not unusual considering the higher number of mutations within these regions compared with highly conserved coding regions (Ebert & Peakall, 2009). Conversely, the number of SSRs in the LSC region was significantly higher than that in the SSC and IR regions, which is consistent with the results of previous research results showing uneven distribution of SSRs in the chloroplast genome (Qian et al., 2013).

Based on the chloroplast genome of M. sieversii as a reference, sequence identification maps of four Malus varieties and two progenitor species were generated. Compared with the IR regions, the LSC and SSC regions showed more differences, and non-coding regions were less conservative than the coding regions. In contrast, similar to the chloroplast genomes of other plants, all rRNA genes were highly conserved (Dong et al., 2013). Comparative analysis showed that different types of DNA fragments had different degrees of sequence variation. In general, the variation in exons was lower than that in introns, and this trend was also observed in other analyses (Cheng et al., 2020). Three variable regions—psbI-atpA, psbM-psbD, and ndhC-atpE—were found, which may be specific DNA barcodes for effective apple variety identification, or potential molecular markers for apple germplasm resources. Single nucleotide polymorphism (SNP) mainly refers to DNA sequence diversity caused by the variation of a single nucleotide at the genomic level, and such polymorphisms are rich and numerous (Liu et al., 2016). in this study, 135 SNP sites were found. SNPs provide an important basis for studying genetic variation in human families, animals and plants. Therefore, they are widely used to study population genetics (including biological origin, evolution, and migration) and disease-related genes.

It is believed that M. sieversii is the primary progenitor, and M. sylvestris is a major secondary contributor to cultivated apple; however, there is no direct and sufficient evidence to this (Nock et al., 2015). Phylogenetic analysis showed that all Rosaceae species formed a large branch with a strong support value. The four cultivated apples were closely related to M. sieversii and M. sylvestris. M. sieversii and Red Delicious formed a cluster, while the Golden Delicious, Red Fuji, Ralls, and M. sylvestris clustered into another branch. Therefore, M. sieversii was not the only parent source of the modern cultivated apple, and M. sylvestris was also a major progenitor. These results are consistent with those reported by Sun et al. (2020). The results provide important theoretical values for the protection and utilisation of the germplasm resources of the genus Malus and contribute to understanding parent selection and breeding of high-yield varieties.

Conclusions

In this study, the chloroplast genomes of four cultivated varieties of Malus were sequenced and compared with the two wild species of Malus. The similarities and differences in the chloroplast genomes of Malus and the genetic relationship between cultivated and wild species were revealed. Intraspecific differences among the chloroplast genomes mainly exist in the intergenic spacer, which is the main feature of the observed genome size differences. In addition, phylogenetic relationships of species in the genus Malus were reconstructed based on whole-genome sequences. The tree showed that varieties of M. domestica and wild apples (M. sylvestris and M. sieversii) formed a branch, which indicated that M. sylvestris and M. sieversii had a close relationship with cultivated apple. This supports the view that M. sylvestris and M. sieversii are the common ancestors of cultivated apples. Furthermore, three cpDNA marker sequences—psbI-atpA, psbM-psbD, and ndhC-atpE—were identified that could be used to study the intraspecific genetic structure and diversity of Malus. This study reveals the relationship between cultivated apples and their possible ancestors based on the molecular analysis of the chloroplast genome. The corresponding molecular data can provide theoretical guidance for the use of apple resources and environment protection.

Supplemental Information

Supplemental Information 1 Genome base composition of apple chloroplast.

WW: Wunan Town, Wuwei City; LX: Xingcheng, Liaoning; ZZ: Zhengzhou Fruit Tree Institute

Click here for additional data file.

Supplemental Information 2 Statistics on four cultivated apple and two wild species SSR loci.

P1: mono-nucleotide; P2: di-nucleotide; C: hexa-nucleotide; LSC: Number of SSR loci in the LSC region; SSC: Number of SSR loci in the SSC region; IR: Number of SSR loci in the IR region.

Click here for additional data file.

Supplemental Information 3 Number of simple repeats in different chloroplast genomes.

Click here for additional data file.

Supplemental Information 4 Supplemental Table S4 Codon usage in six Malus chloroplast genomes.

Click here for additional data file.

Supplemental Information 5 SNP data analysis of Malus.

WW: Wunan Town, Wuwei City; LX: Xingcheng, Liaoning; ZZ: Zhengzhou Fruit Tree Institute

Click here for additional data file.

Supplemental Information 6 InDels data analysis of Malus.

Click here for additional data file.

Supplemental Information 7 Experimental data.

Click here for additional data file.

Additional Information and Declarations

Competing Interests

Author Contributions

Data Availability

The authors declare that they have no competing interests.

Xueli Li performed the experiments, analyzed the data, prepared figures and/or tables, authored or reviewed drafts of the paper, and approved the final draft.

Zhijie Ding performed the experiments, prepared figures and/or tables, and approved the final draft.

Haoyu Miao performed the experiments, authored or reviewed drafts of the paper, and approved the final draft.

Jinbo Bao performed the experiments, prepared figures and/or tables, and approved the final draft.

Xinmin Tian conceived and designed the experiments, authored or reviewed drafts of the paper, and approved the final draft.

The following information was supplied regarding data availability:

The raw measurements are available in the Supplementary File.

The sequences are available at GenBank: OK458681, OK539527, OK539528, OK539529, OK539530, OK539531, OK539532, OK539533, OK539534, OK539535, OK539536, OK539537, OK539538, OK539539, OK539540, OK539541, OK539542, OK539543, OK539544, OK539545, OK574627, OK574628, OK574629, OK574630.

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
