# Peer review of "Complete chloroplast genome studies of different apple varieties indicated the origin of modern cultivated apples from Malus sieversii and Malus sylvestris"

_PeerJ, doi:10.7717/peerj.13107_

## Round 0.1 · original submission · Major Revisions

As you can see, all three reviewers found some issues and recommended major revision. Please address all the concerns of the reviewers and amend your manuscript accordingly.

Reviewer 1 ·

Basic reporting

The manuscript needs professional language editing.

Experimental design

The experimental design is too simple to answer the question raised by the authors.

Validity of the findings

The major conclusion is neither original nor well supported

Additional comments

Li and colleagues assembled the chloroplast genome of 24 cultivated apple accessions and characterized the features and phylogeny of the genomes. In the past decade, numerous studies have been published to dissect the origin and domestication of apple using both morphological and genomic evidence. Presently, one consensus reached by the community is that Malus seiversii and Malus sylvestris are the two major wild progenitors of modern cultivated apples. This conclusion has been supported by a wide array of population-scale studies on both nuclear and chloroplast genomes, including Nikiforova et al Mol Biol Evol 2013, Duan et al Nat Commun 2017, and Sun et al Nat Genet 2020. It should be noted that some early cultivars as well as the specific local cultivars may have been originated from different wild species (Chen et al. Plant Biotechnol J 2021). However, cultivars used in this study are presumably the descendant of sieversii and sylvestris, which is consistent with finding of this work from small scaled chloroplast genome analysis. In summary, the authors have used a small-scaled dataset to explore a big and largely answered question, therefore I don’t think this study provides new information for the apple research community. By the way, figures in this manuscript are not easy for visualization, which should be revised for future publication.

Specific points:
Line 16: change “in this world” to “in this work”
Line 43 formatting error for “__world”
Line 46-49, the statements are incorrect, there are substantial studies addressing the origin and domestication of cultivated apples, e.g. Cornille et al. 2012 & 2014, Duan et al. 2017, Sun et al. 2020.
Line 66-67, according to the recently published studies cited above, there is an agreement on M. sieversii and M. sylvestris being as the major genetic contributors of modern cultivated apples, however, the relative contribution of the two wild progenitors to cultivated apples varies depending on varieties.
Line 72-73, shouldn’t the nuclear genome be more important than the organelle genomes?
Line 98-100, again, much of this has been addressed in previous studies.
Line 111, I don’t understand why the authors sampled multiple individuals of the same cultivar, since apple tree is propagated through grafting, different individuals should be genetically similar.
Line 117 these are not 24 “varieties”, they are 24 “accessions”.
Line 120, there are apple chloroplast genomes published already, why did the authors choose Pyracantha fortuneana as the reference?
Line 172-175, accession numbers should be provided.
Line 270, it is inaccurate to describe it as a “new branch”

Reviewer 2 ·

Basic reporting

The manuscript "Complete chloroplast genomes studies of different apple varieties indicated modern cultivated apples originate from Malus sieversii and Malus sylvestris" submitted to PeerJ journal. In this study, complete chloroplast genomes of 24 apple varieties by Next-generation sequencing technology were obtained, and compared with the wild species M. sylvestris and M. sieversii. The similarities and differences of chloroplast genomes of Malus and the genetic relationship between cultivated varieties and wild species were revealed. Several mutation hotspot regions were detected, and phylogenetic analysis revealed that M. sylvestris and M. sieversii were the ancestor species of cultivated apple. These results provide effective information for solving specific molecular markers in taxonomic research, also provide a certain reference value for resources protection and new varieties breeding. The manuscript is clear, substantial and well organized. However, I have some concerns, which need to be addressed before considering being accepted for publication.

1. In the part of Materials & Methods, why this paper used Pyracantha fortuneana (MH890570.1) as reference using the software NOVOPlasty, whereas not use M. sylvestris (MK434924.1) or M.sieversii(MK434920)?
2. In line 120-121, please describe the annotation software used in detail. In general, Sequin is not an annotation software.
3. In line 141, Please describe parameters of MISA software.
4. In line 162-168, to identify single nucleotide polymorphism (SNP) or short insertion/deletions (indels), you used which as a reference genome?
5. In line188, please replace “160067-160069” with “160067-160069 bp”.
6. In line192-193, the author said “M.sieversii and M. sylvestris contained 128 genes, including 83 CDS genes, 35 transfer RNA and eight ribosomal RNA genes.”. However, 83 CDS genes +35 transfer RNA +8 ribosomal RNA genes =126 genes. The same error appears on lines 193-194. Please check it.
7. In line 210, The rpl16 gene should be italicized.
8. In line 260, “non coding region” should be unified.
9. Almost all Figures are not clear, except for figure 2 and 7, please change the picture with a higher definition.

Experimental design

no comment

Validity of the findings

no comment

Additional comments

no comment

Reviewer 3 ·

Basic reporting

English/text : The ms is overall difficult to read. Work to streamline the ms is needed to understand the questions asked and in particular data used. The introduction should better reflect the state of the art (at least remove the contradictory ideas developed in different paragraphs, see comments below).

Figures :
-The figures are numerous (7!) and some of them are difficult to read on my computer (e.g. Figure 1 and 5).
-the legend should be self-explanatory (e.g. avoid abbreviations, e.g., in Figure 2 “LSC, SSC and IR” or Figure 7, precise which “Bayesian” inference was used)
-sometimes it is complex to know with which data the Figures were generated, to be precised

Tables: Tables should be self-explanatory, e.g.
Table 1: put the meaning of each header below the table (e.g. what does mean “dimension”?)
Table 2: put the meaning of each header below the table or in the legend (e.g. what does mean “LSC”)?
Tables 3 and 5: could be moved in supplementary material.

Experimental design

The research question is meaningful, but the authors explained that there is an identified knowledge gap on the contribution of wild apple species to the cultivated apple genome, which I do not fully agree with, see my comments below. In addition, the samples used in this study are not fitting to test the hypotheses the authors would like to test. Indeed, using one reference sample of each of the two wild contributors is not sufficient to make solid conclusions on the contributions of wild species to crop genomes. Turning the ms/hypothesis towards the comparative genomics of chloroplast genome among wild and cultivated apples should be more appropriate. Eventually, I found the sequencing method and the way in which they “assembled” the genomes (if it is what has been done?) unclear.

Validity of the findings

Because of the comments above I cannot validate or not the findings.

Additional comments

Li, Ding et al compare the genome structures of the chloroplast (cp) of Malus sieversii and M. sylvestris, the two main wild contributors to the cultivated apple genome (Malus domestica), with the cp genomes of four cultivated apple varieties. Albeit I found the question interesting, I couldn’t understand what the authors tried to communicate in several parts of the ms. See below my comments. Some part of the methodology is often unclear and sequencing technology and methods to extract the sequence information (assembly? Or mapping?) used are not described. I therefore cannot agree or not with the conclusions of the authors as I miss information.


*MAJOR COMMENTS

-I couldn’t understand what the authors tried to communicate in several parts of the ms, please see comments below;
-The authors talk about “Next generation sequencing” without explaining clearly the technique used. Along the ms, it is not clear which data they generated, and which data they retrieved, see comments below;
-In the introduction the authors insist on the fact that there are still debates on the contributions of different wild apple species (initial domestication or hybridisation, lines 46-50 and 66-67) to the cultivated apple genome, but I do not fully agree with: there are a lot of consistent evidences that show that M. sieversii is the initial ancestor, and M. sylvestris has contributed through recent wild-to-crop introgressions.
-In addition, the samples used for answering the question are not appropriate, they used four cultivars of M. domestica (and 1 to 2 repeats per cultivar from different locations, Table 1), and only two already available genomes of sieversii and sylvestris. In addition, the authors do not show how the use of data (M. sieversii and M. sylvestris) that were not generated with the same sequencing techniques as their genomes (I presume?), could impact (or not) their results.

*MINOR COMMENTS

Line 39: correct “Goldie Delicious” by“Golden Delicious”
Lines 46-50 : this sentence sounds surprising, we now have a good idea of the origin of the apple and the role of the different wild species (see Duan et al 2017, Sun et al. 2021, Cornille et al 2012, Migicovsky 2020). This sentence also stands in contrast with the next paragraph. Please rephrase and add citations.
Lines 66-67: this statement sounds in contradiction with the previous sentence explaining what is known about the contribution of different wild apple species to the cultivated apple genomes. Please rephrase.
Line 79: the authors use “cp” from there in the ms, but never explain this term as referred to as “chloroplast”. This should be clearly stated.
Line 98 “So who are the wild ancestors of cultivated apples in the process of the origin and evolution of cultivated apples? Is the origin of cultivated apple direct domestication or hybrid origin?” The authors cannot answer specifically to this question only using cp genomes, and with only one sample of each wild contributor. I would remove or rephrase this sentence. In addition, some studies already investigated these questions, so the authors should adapt their questions regarding previous work done.
Line 103 : “In this study, 24 genomic sequences of four different Malus germplasm were deeply sequenced.” Unclear, what do the authors mean by “deeply” sequence?
Line 104 “Through the detection of DNA variation at the genomic level” pleonasm, remove “at the genomic level”
Line 118: Detailed the sequencing techniques, how the libraries were prepared, and DNA extract? What are the differences in the way in which the M. sieversii and M. sylvestris genomes were generated?
Line 120: precise why Pyracantha fortuneana has been used as reference.
Line 125: the authors used a lot “LSC, SSC, IRS” as abbreviations for the three regions, but did not introduce them before. They should be introduced before for non experts of chloroplastic genomes.
Line 155: Variation analyses of what ? Precise
Line 156: a word is missing after “conserved”, precise
Line 170 “this cp genome phylogenetic placement in the Malus genus”, weird wording, please rephrase.
Line 177: precise which “ Bayesian method”, and explain quickly the method for non-expert (and if you used MCMC, precise the burin and number of MCMC steps, and additional basic parameters).
Line 193: “in cultivated apple,”, unclear, in the four cultivars you mean? Precise
Line 102 to 205: this paragraph is very descriptive and redundant with Table 2? To shorten?
Line 268: remove “that the evolutionary tree had”
Line 278: “assembled”: really? I do not see how the assembly was performed? Explanations in the “methods” part are needed.

I did not continue beyond the “results” part, as I need more clarification in Mat&Meth to go further.

*NOTES ABOUT English/grammar typos (they are likely not exhaustives);
Line 50-51 “To have a better comprehends the origin of cultivated apple and its relationship with main wild resources, many”, correct the sentence
Line 55: “howere” correct by “However”
Line 60: “contended that The”, remove capital letter
Line 68-73: this sentence needs to be correct for English + grammar (“genome level traced ancestry”, “it”, to what refers to “it”: unclear, researches without “es”…)
Line 83 “geneme” correct by “genomes”
Line 84 correct “have” by “has”
Line 86: problem in the author’s name “SSvetlana”
Line 88: miss a space between apple and M. “apple(M. sieversii”
Line 89 “the contribution of wild species to domesticated maternal contribution” unclear sentence
Line 92-93: “In addition, due to complete cp genome analysis can provide more genetic information than gene fragments” Pb in the sentence?
Line 98: correct “who” by “which”
Line 135: add a space between repeat and SSRs “s(‘
Line 144: remove the dot to the title, and U in lower case
Line 187: put results in the past tense
Line 192: add a space between M. and sieversii
Line 271: remove “It was obvious”
Line 272: correct “Goldie” by “Golden”.
Along the ms : problem with the references : there is often a space missing between the reference in bracket and the text.

---

## Round 0.2 · Minor Revisions

Please address remining issues indicated by the reviewer and amend your manuscript accordingly.

Reviewer 1 ·

Basic reporting

The revised manuscript has been much improved on both languish and content. Most of my previous concerns have been addressed and I support the publication of this manuscript on PeerJ as long as the typos are corrected (see below for comments).

Experimental design

no comment

Validity of the findings

no comment

Additional comments

Line 82, change "the genome of" to "the genome and"

Line 137, please correct the typo "genemes"

Line 148, change "A-tail" to "polyA-tail"

Line 149, change "sequencing junction" to "sequencing adapters"

Line 236, correct the typo "consiststed"

---

## Round 0.3 · accepted · Accept

All remaining issues pointed out by the reviewer were adequately addressed and the manuscript was revised accordingly. The amended version is acceptable now.